

# The role of ecosystem engineers in shaping the diversity and function of arid soil bacterial communities

Capucine Baubin[1], Arielle M. Farrell[2], Adam Šťovíček[3], Lusine Ghazaryan[1], Itamar Giladi[2], Osnat Gillor[1]

[1]Zuckerberg Institute for Water Research, Blaustein Institutes for Desert Research, Ben-Gurion University of the Negev, Israel
[2]The Mitrani Department of Desert Ecology, Swiss Institute for Dryland Environmental and Energy Research, Blaustein Institutes for Desert Research, Ben-Gurion University of the Negev, Israel
[3]Department of Microbiology, Nutrition and Dietetics, Czech University of Life Sciences Prague, Kamycka 129, Prague 6, 16500, Czech Republic

*Correspondence to:*

Capucine Baubin, Zuckerberg Institute for Water Research, Blaustein Institutes for Desert Research, Ben-Gurion

University of the Negev, Israel. Tel: 972-54-2944886; e-mail: baubin@post.bgu.ac.il

Osnat Gillor, Zuckerberg Institute for Water Research, Blaustein Institutes for Desert Research, Ben-Gurion

University of the Negev, Israel. Tel: 972-8-6596986; e-mail: gilloro@bgu.ac.il



**ABSTRACT**
Ecosystem engineers (EEs) are present in every environment and are known to strongly influence
ecological processes and thus shape the distribution of species and resources. In this study, we
assessed the direct and indirect effect of two EEs (perennial shrubs and ant nests), individually and
combined, on the composition and function of arid soil bacterial communities. To that end, topsoil
samples were collected in the Negev Desert Highlands during the dry season from four patch types:
(1) barren soil; (2) under shrubs; (3) near ant nests; or (4) near ant nests situated under shrubs. The
bacterial community composition and potential functionality were evaluated in the soil samples
(fourteen replicates per patch type) using 16S rRNA gene amplicon sequencing, together with
physico-chemical measures of the soil. We have found that the EEs differently affected the community
composition. Barren patches supported a soil microbiome, dominated by *Rubrobacter* and
*Proteobacteria*, while in EE patches *Deinococcus-Thermus* dominated. The presence of the EEs
similarly enhanced the abundance of phototrophic, nitrogen cycle and stress- related genes. In
addition, the soil characteristics were altered only when both EEs were combined. Our results suggest
that arid landscapes foster unique bacterial communities selected by patches created by each EE(s),
solo or in combination. Although, the communities' composition differs, they support similar potential
functions that may have a role in surviving harsh arid conditions. The combined effect of the EEs on
soil microbial communities is a good example of hard to predict non-additive features of arid
ecosystems that, therefore, merit further research.



## 1. INTRODUCTION

Hot desert environments are characterized by long droughts interspersed by intermittent and
unpredictable rain events. Water and nutrients in hot desert environments are scarce and unevenly
distributed across the land, resulting in patches of contrasting productivities. High-productivity
patches, also called resource islands, are defined by large concentrations of organic matter and
nutrients (Ben-David et al., 2011; Schlesinger et al., 1996; West, 1981). These resource islands can be
formed through the redistribution of nutrients and water by ecosystem engineers (EEs), such as
perennial plants or invertebrates (Wilby et al., 2001; Wright et al., 2006). EEs are also known for
impacting many components of a given environment, such as soil features, annual distribution, or
community composition of microorganisms (De Graaff et al., 2015; Oren et al., 2007).

An EE is an organism that, directly or indirectly, modifies the availability of resources to other
organisms by transforming the physical state of abiotic and/or biotic components of the ecosystem,
*sensu* Jones et al. (1994). The impacts of EEs range from physical, through the creation of biogenic
structures (e.g. tunnels) (Lavelle, 2002); to chemical, through the production of compounds that have
physiological effects (e.g. root exudates) (Lavelle et al., 1992); to biological, through organisms
behaviour (e.g. seed dispersal) (Lavelle et al., 2006). In drylands, resources, such as nutrients or water,
are often concentrated around EEs, boosting the development of diverse populations of annual plants
and invertebrates (Wright and Upadhyaya, 1996), as well as microbial communities (Bachar et al.,
2012; Ginzburg et al., 2008; Saul-Tcherkas and Steinberger, 2011). The community's taxonomy is
linked to its' potential function (Narayan et al., 2020), responding to the physico-chemical conditions.
This implies that the variation in taxonomy by the presence of an EE would be associated with
changes in potential function.
In desert ecosystems, ants are a notable example of an EE (Ginzburg et al., 2008). They redistribute
resources by tilling the soil, bringing soil from the deep layers to the upper layers (bioturbation), and
by gathering, storing, and ejecting food items, such as plant material, or dead invertebrates, in and
around the nest (Filser et al., 2016; Folgarait, 1998; MacMahon et al., 2000). EEs in arid environments



also include perennial shrubs (Callaway, 1995; Schlesinger and Pilmanis, 1998; Segoli et al., 2012;
Shachak et al., 2008; Walker et al., 2001). Their root systems create a soil mound that traps litter, and
seeds, allowing for higher water infiltration. The root exudates increase the content of organic matter
and the shrub canopies decrease evaporation, prolonging water availability following a rain event
(Bachar et al., 2012). In addition, the presence of shrubs alters the course of water run-off (Oren et al.,
2007), which impacts the locations of available water for soil microbial communities. In addition, root
systems have their own microbiome, which interact with the soil microbial community (Steven et al.,

74 2014).

The role of both ants and perennial shrubs as EEs were reported in various ecosystems (Facelli and
Temby, 2002; Farji-Brener and Werenkraut, 2017; Frouz et al., 2003; Gosselin et al., 2016; Pariente,
2002; Schlesinger et al., 1996). However, we know little about their joint effect in arid ecosystems.
We hypothesized that each EE would shape a unique soil bacterial community via changes in the soil
physico-chemical properties. We further predicted that since shrubs canopy and ant nests may
differently affect soil properties, their combined effect on the microbial community is non-additive
and thus cannot be predicted by the contribution components. To test our hypotheses, we explored arid
soil bacterial microbiomes and soil chemical features during the dry season of 2015. We sampled four
different patches: under *Hammada scoparia* shrubs; near the nest openings of the harvester ant,
*Messor ebeninus*; in combined patches of nests under shrubs; and in barren soil.





## 2. MATERIALS AND METHODS

### 2.1. Sampling

The study was conducted in a long-term ecological research (LTER) site in the Central Negev Desert, Israel (Zin Plateau, 34°80'E, 30°86'N). It is characterised by a 90 mm annual rainfall and average monthly temperatures fluctuating from 13°C (January) to 35°C (August). Vegetation is scarce and dominated by the perennial shrubs *Hammada scoparia* and *Atriplex halimus* (Gilad et al., 2004).

Sampling was conducted as previously described (Baubin et al., 2019) with slight modifications, such as the inclusion of Shrub&Nest samples. To summarize, we sampled four distinct patch types: (1) barren soil (Barren); (2) under the canopy of *H. scoparia* (Shrub); (3) 20-30 cm from the main opening of the nest of *M. ebeninus* (Nest); and (4) 20-30 cm from ant nest's opening that was situated under a shrub canopy (Shrub&Nest). Samples were collected in October 2015, after an eight-month drought.

We sampled 14 random experimental blocks, from each of the four patches (4 patch types x 14 blocks = 56 samples). All samples were collected from the top 5 cm of the soil after removing crust and debris, then processed within 24 hours of collection. In the lab, the soil from two adjacent blocks was composited resulting in 28 samples that were further processed. Each sample was sieve-homogenized through a 2 mm mesh. 5 g of soil were stored in -80°C for molecular analysis, 20 g were used for water content analysis and the rest was dried at 65°C and used for physico-chemical analysis.

### 2.2. DNA extraction, amplification, and sequencing

Total nucleic acids were extracted from 0.5 g of soil as previously described (Angel, 2012), purified with the ExgeneTM Soil SV kit (GeneAll, Seoul, S. Korea) according to the manufacturer's instructions. The 16S rRNA encoding genes V3-V4 region was amplified using 341F and 806R primer (Klindworth et al., 2013). The PCR reaction consisted of 2.5 µL 10x standard buffer, 10 µM primers, 0.8 mM dNTPs, 0.4 µL DreamTaq DNA polymerase, 4 µL template, 1 mM bovine serum albumin (Takara, Kusatsu, Japan) and 12.6 µL Milli-Q water. Triplicate PCR reactions (95°C for 30 secs; 28 cycles of 95°C for 15 secs, 50°C for 30 secs, 68°C for 30 secs; 68°C for 5 min) were pooled and



amplicon concentration and purity were measured by electrophoresis, Nanodrop (ND-1000, Thermo
Fisher Scientific, Waltham, MA, USA). The amplicon libraries were constructed and sequenced on the
Illumina MiSeq platform (2x250 pair-end) at the Research Resources Centre at the University of
Illinois.

### 2.3. Soil physico-chemical analysis

The physico-chemical parameters of the soil samples were assessed following the standard methods
(SSSA, 1996). Water content was measured by gravimetry. Other parameters were measured as
follows by the Gilat Hasade Services Laboratory (Moshav Gilat, Israel): organic matter (OM) content
by dichromate oxidation; nitrate ($NO_3^-$) through aqueous extract; ammonium ($NH_4^+$) through KCl
solution extract; phosphorus (P) by sodium bicarbonate extract; and pH in saturated soil extract. The
soil parameters were plotted using a Principal Component Analysis (PCA) (stats package (R Core
Team, 2016)) and the significance of difference between patches was evaluated using a non-
parametric test: Kruskal-Wallis test and a post-hoc Dunn test (Dinno, 2017; Dunn, 1964; Kruskal and
Wallis, 1952).

### 2.4. Community analysis

The results were analysed using QIIME2 (Bolyen et al., 2018) and Dada2 (Callahan et al., 2016),
following the NeatSeq-Flow pipeline (Sklarz et al., 2018) and Amplicon Sequence Variants (ASVs)
were created. The taxonomic assignment was done using Silva (version 132) (Quast et al., 2013),
through QIIME2 and the statistical analysis was done using R (R Core Team, 2016). A NMDS plot
was created using the Bray-Curtis dissimilarity and the significance of differences between patch types
was analysed using ANOSIM (vegan package (Oksanen et al., 2014)). The taxonomy was plotted
using a stacked bar plot and the significance of difference between patch types was assessed using a
non-parametric test: Kruskal-Wallis test and a post-hoc Dunn test (Dinno, 2017; Dunn, 1964; Kruskal
and Wallis, 1952). All sequences retrieved in this study were uploaded to BioProject
(https://www.ncbi.nlm.nih.gov/bioproject) under the submission number PRJNA484096.



## 2.5. Functional Prediction


The prediction of function of the 16S amplicons was done with Piphillin using the KEGG database
(October 2018). Piphillin generates a genome abundance table that is normalized to the 16S rRNA
copy number for each genome (Iwai et al., 2016; Narayan et al., 2020). To analyse the arid soil
microbial functionality, we selected metabolisms and respective genes related to arid soil using groups
and genes from the KEGG database (Kaneshisa and Goto, 2000). We selected steps in metabolic
pathways for different methods of harvesting energy (organotrophy, lithotrophy and phototrophy)
(Cordero et al., 2019; Greening et al., 2016; León-Sobrino et al., 2019; Tveit et al., 2019), for parts of
the nitrogen cycle (Madigan et al., 2009), and for the survival of the individual during a drought (DNA
conservation and repair, sporulation and Reactive Oxygen Species (ROS)-damage prevention)
(Borisov et al., 2013; Hansen et al., 2007; Henrikus et al., 2018; Preiss, 1984; Preiss and Sivak, 1999;
Rajeev et al., 2013; Repar et al., 2012; Slade and Radman, 2011). Then, we looked for each step in the
KEGG database and picked out genes of interest to build our own database. The assignment of
function to the KEGG numbers was done in R. The significance of the differences between patch
types in predicted functionalities was evaluated using a non-parametric test: Kruskal-Wallis test and a
post-hoc Dunn test (Dinno, 2017; Dunn, 1964; Kruskal and Wallis, 1952) and boxplots were created in
R.





## 3. RESULTS

### 3.1. Soil physico-chemical characteristics

The PCA (Figure 1) depict differences in the soil characteristics (listed in Table A1) between the

Shrub&Nest and the other patches (barren, nest, and shrub). Therefore, we will present the average of

these other patches compared to the Shrub&Nest average. The variance of the data is explained to

99.6% by the two first principal components. The difference between patches is driven by the high

concentrations of $NO_3$ (4.7 mg/kg compared to 30 mg/kg, respectively) and P (22 mg/kg compared to

54 mg/kg, respectively). When verifying with a Kruskal-Wallis test and a Dunn test on the values of

these soil variables (Table A2), we see that the differences between patch types are significant

(Shrub&Nest vs all other patches, $p < 0.05$). Patches with two EE also have a significantly higher

concentration of $NH_4$ (9.72 mg/kg) and OM (8.21%) compared to all other patches ($NH_4$ mean: 5.62

mg/kg, p-value $<0.05$; OM mean: 5.51%, $p \leq 0.05$). However, the water content and pH did not show

significant differences between patches (Table A2).

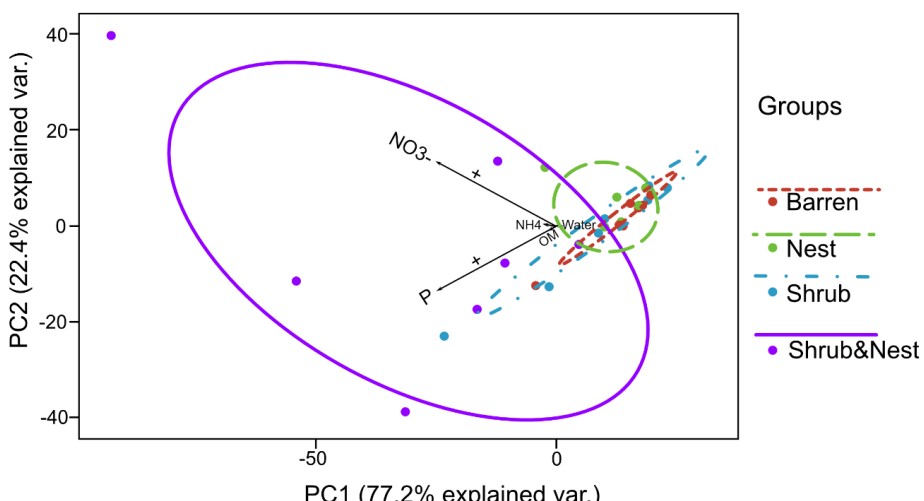

Figure 1. Principal Component Analysis of the soil parameters (NO3= Nitrate , P = Phosphorus, NH4

= Ammonium, OM = Organic Matter content, Water = Water content). The plus signs on the $NO_3^-$ and

the P vector show an increase in concentration in the Shrub&Nest patches.





### 3.2. Beta diversity


The summary of the sequence analysis can be found in Table A4. DADA2 analysis yielded 2318
ASVs and the NMDS results (Figure 2) suggests that there are significant differences in the microbial
community between patch types (ANOSIM, R= 0.28; p = 0.001). Most notably, the barren soil
microbial communities (red circles) that were sampled in barren soil patches showed high similarities
between blocks and were significantly different (p < 0.05, Table A3) from the communities of other
patch types (high clustering of barren soil sampling points in the NMDS space). In contrast, the
dissimilarities in community composition within the patch types that included shrubs (Shrub and
Shrub&Nest) were high (large scatter of sampling points in the NMDS space).

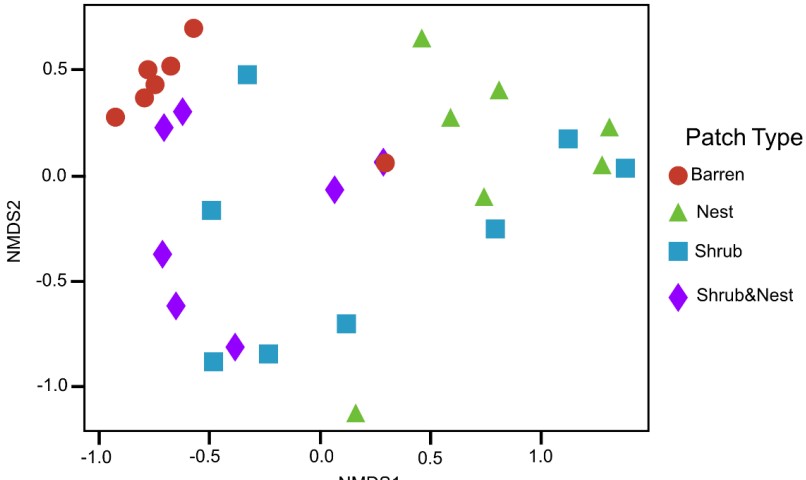


Figure 2. Non-Metric Multidimensional Scaling (NMDS) of the soil 16S microbial communities in the
dry season under different patch types. The patch types are significantly different from each other
(ANOSIM, R= 0.28247; p-value = 0.001)

### 3.3. Community composition


The community was mostly composed of *Actinobacteria, Proteobacteria, Deinococcus-Thermus,*
*Bacteroidetes* and *Firmicutes* (Figure 3). The relative abundance for each phylum is detailed in Table
A5. We focused on the results of the main three phyla: *Actinobacteria, Deinococcus-Thermus* and





*Proteobacteria*. Using pair-wise comparisons, we saw that shrub patches and nest patches had similar
communities (no significant differences, p > 0.05) therefore, we considered them as single EE patches.
For these patches, an average relative abundance of nest and shrub patches was used for statistical
data. For the *Actinobacteria* phylum, patches with one EE had significantly lower relative abundance
than barren patches (one EE: 9 % vs barren patch: 35% p < 0.005) or patches with two EEs (17%, p-
value: 0.02). For the *Deinococcus-Thermus* phylum, barren patches had significantly lower relative
abundance than patches with one or two EEs (Barren: 3%; vs one EE: 25%; vs two EEs: 9%, p <
0.05). A similar pattern was detected in the *Proteobacteria* phylum (Barren: 38%; vs one EE: 44%; vs
two EEs: 39%, p < 0.05). All p-values can be found in Table A6.

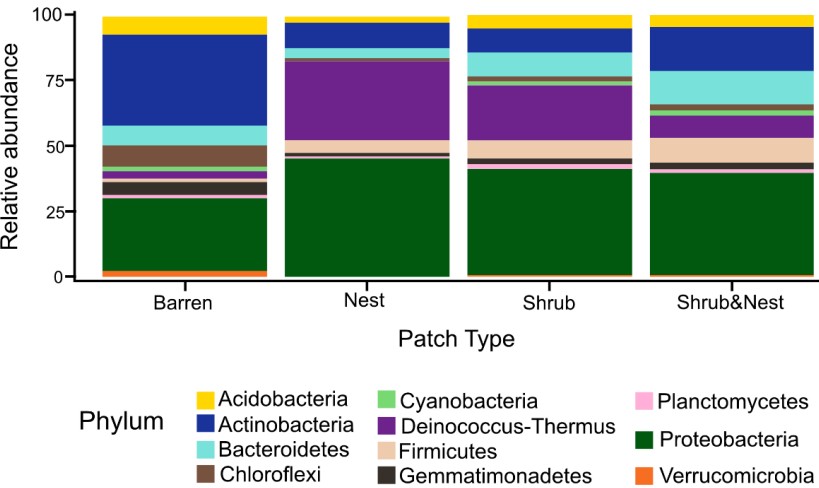


Figure 3. Barplot of the relative abundance (in %) of the most abundant phyla in the soil microbial
community in the dry season under different patch types (phyla with a relative abundance > 0.05%).
The relative abundance of *Deinococcus-Thermus* increases when one EE is present while the
population of Actinobacteria decreases.





### 3.4. Functional prediction

The abundance of each gene group has been normalized to the 16S rRNA copy number for each genome. The functional prediction results focus on eight distinct gene groups: Phototrophy, Lithotrophy, Organotrophy, DNA Conservation, DNA Repair, Nitrogen cycle, Sporulation and ROS-damage prevention (listed in Table A7). Figure 4 shows the pattern of the obtained functions. It shows higher abundances of the gene groups encoding for DNA conservation, DNA repair, nitrogen metabolism, ROS-damage prevention, sporulation, and phototrophy in patches associated with at least one EE compared to the barren patches (Table A8). Therefore, we analysed the results as barren vs average of the other three patch types that were not significantly different from one another (Table A9) and significant differences ($p < 0.04$) between barren and EE(s) patches were detected. The genes related to lithotrophy, only differed between patches with one EE and the barren patches ($p < 0.03$) but patches with two EEs were similar to the barren plots. Finally, for genes related to the organotrophy, there was no significant differences between the patches ($p > 0.05$).





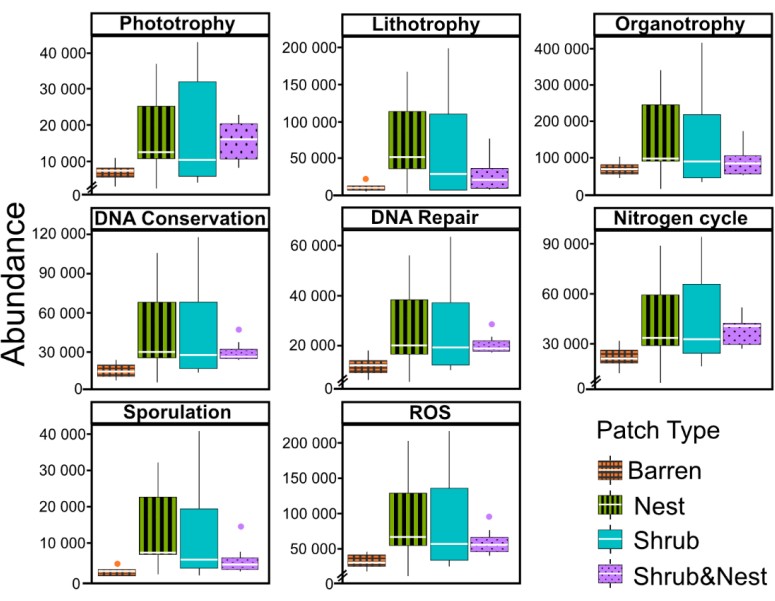

Figure 4.


Figure 4. Boxplots of the functional prediction of the 16S sequences. Each panel (Boxplot) represents
a different group of genes associated with a certain functionality. The full list of genes can be found in
Table A7. The patch types are represented by distinct colours and patterns. The y-axis is the
abundance in copy number (CN) normalized to the 16S rRNA copy number for each genome.



## 4. DISCUSSION


In desert environments, during the dry season, a large portion of the microbial community is dormant
or showing reduced metabolic activity (Bay et al., 2018; Cordero et al., 2019; Lennon and Jones,
2011; Schulze-Makuch et al., 2018). However, the presence of EEs enhances the metabolic potential
for metabolism-related and the survival-related functions (Figure 4). This implies that the soil
microbial communities occupying EE patches are better adapted to confront stressful events (e.g.,
sudden rewetting or desiccation). However, these communities experience more habitable conditions
due to the modulating effects of the EEs on the environmental conditions. The increase in the activity
of gene groups can be explained by an increase in nutrients in the joint EEs patches (Table A1).
However the physico-chemical measures, including soil water content, OM, nitrogen, P, and pH, did
not match the changes observed in bacterial composition or function (Table A1, A2, A9 and Figure 1)
as was previously reported (Angel et al., 2010; Bachar et al., 2012; Vonshak et al., 2018). Indeed,
there was no significant link between the changes in the bacterial communities and the soil parameters
(Table A10). We have previously proposed that the observed differences in communities could be
mediated by microclimatic characteristics under shrub patches (Bachar et al., 2012). It has been
reported that the desert dwarf shrubs affect the physical features of their immediate soil patch. Shrubs
were shown to divert water flow and reduce evapotranspiration rates following rain events (Sarig and
Steinberger, 1993; Segoli et al., 2008; Whitford and Duval, 2002) and reduce temperature and
radiation year round (Kidron, 2009). Likewise, ants aerate the soil thus increasing infiltration during
rain events (Berg and Steinberger, 2008) and mix the layers through bioturbation (Folgarait, 1998).
Therefore, the prolonged water availability and altered physical conditions from the wet season may
hold lasting effects on the community structure (Baubin et al., 2019), establishing the composition and
functions observed here (Figure 3 and 4).
Both *Actinobacteria* and *Deinococcus-Thermus* were abundant in all patches, but their relative
abundance was negatively correlated. Their two dominant genera are both well adapted to stress
conditions: *Rubrobacter* dominated the barren soil, while *Deinococcus* dominated the EE patches
(Figure 3 and Table A5). *Rubrobacter* are specialized in surviving strong desiccation and low nutrients



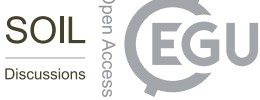

(Bull, 2011; Ferreira et al., 1999) showing high relative abundance in arid barren soils of the Negev
highlands (Meier et al., 2021). *Deinococcus* are highly adapted to a wide range of extremes, such
radiations, temperatures and, xerification. Some of these extreme conditions occur in the desert, while
others are found in different environments, making *Deinococcus* versatile organisms (Chanal et al.,
2006; Prieur, 2007; Slade and Radman, 2011). This versatility allows them to thrive in EE patches as
they can better adapt to perturbations compared to *Rubrobacter*.
Only the combination of EEs resulted in significant changes (p-values: Table A2) of $NO_3^-$, P, and, to a
lesser extent, $NH_4^+$, pH, and OM (values: Table A1). When located under a shrub, ants can increase
their seed consumption, which enhances the amount of leftovers around the nest (Wagner, 1997), and
increase the concentrations of $NO_3^-$ and P. These macronutrients are important drivers of the biological
processes, as they are often the limiting factors of microbial growth and activity in the terrestrial
environments (Madigan et al., 2009). The EE patches analysed in this study share the same habitat and
resources but their impacts are distinct (Passarelli et al., 2014), thus their joint impact is non-additive.
The impact of an EE is defined by its lifetime, its population density, its spatial distribution, the time
period of its presence on the site, the durability of its impact in the absence of other EEs, and the
number, type, and magnitude of resource flows that are modified (Jones et al., 1994). The behaviour of
each EE is important as it becomes a feature of the combined impact of both EEs (Alba-Lynn and
Detling, 2008). However, the effect of both EEs together cannot be inferred from their individual
environmental impact or from their mutual interaction (Gilad et al., 2004). Here, we investigated a
sessile organism with a passive and slow impact (the perennial shrub) and compared it to a motile
organism (the ants) with an active and transient impact. Ants have both a short-term impact through
the seasonal accumulation of seeds and organic matter and a lasting impact due to the alternation of
the nest mound which remains in the same place for decades (Wagner and Jones, 2004). Even though
their impacts are clearly separated, they create favorable conditions increasing the activity of the
subsoil bacterial communities (Figure 4). Indeed, they create havens of resources and water, which can
be affiliated to the concept of resource islands (Schlesinger and Pilmanis, 1998). However, their



individual, and combined, effects do not always lead to strong changes in the composition of the soil
microbial community (Figure 3).
In our ecosystem, shrubs and ants are not the only two EEs and further studies should also consider the
impact of other EEs. For example, the soil crust and the cyanobacteria living in it are recognized as an
important EE in arid ecosystems (Eldridge et al., 2010; Gilad et al., 2004; Jones et al., 1994; West,
1990). Furthermore, the soil crust in our system is often disturbed by the action of the other two EEs
(Li et al., 2014; Oren et al., 2007). Thus, this third type of EE is not only important for its potential
impact on the microbial community composition and soil physico-chemical properties (Schulz et al.,
2016), but its distribution is also dependent on those of the other two EEs. Such complicated
relationships may explain some of the discrepancies presented in our study.
**5. CONCLUSIONS**
In conclusion, the main stress-resistant phyla (Actinobacteria and Deinococcus-Thermus) react
differently to the presence of EEs. The presences of these EEs also lead to a higher potential activity in
the microbial communities. However, even though they have similar impacts, when together, EEs
have non-additive effects.











**ACKOWLEDGMENTS**

This study was supported by the Koshland Foundation to Itamar Giladi and Osnat Gillor.

**DATA AVAILABILITY**

The data (raw reads) are available in Bioproject under the submission number PRJNA484096.

**COMPETING INTERESTS**

The authors declare that they have no conflict of interest.

**AUTHORS CONTRIBUTIONS**

IG, OG and AMF conceptualized and designed the methodology; AMF and AS collected the samples
and metadata; LG and AMF did the laboratory work and sequencing; CB did the formal analysis,
visualization, data curation and wrote the manuscript; IG, OS and CB did the reviewing and editing of
the manuscript.





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



**APPENDICE A**
Table A1. Soil characteristics data. $NH_4^+$ and P show the highest discrepancy between Shrub&Nest
patches and the other three types.

| ID | pH | NH$_4^+$ (mg/kg) | NO$_3^-$ (mg/kg) | Water content (%) | Organic Matter (%) | P (mg/kg) |
|---|---|---|---|---|---|---|
| Barren | 7.9 | 6.2 | 6.0 | 1.5 | 1.5 | 42.1 |
| Barren | 8.1 | 6.9 | 1.8 | 1.8 | 0.3 | 20.3 |
| Barren | 8.3 | 4.6 | 2.7 | 1.5 | 0.4 | 20.8 |
| Barren | 8.1 | 4.1 | 2.0 | 1.6 | 0.5 | 14.6 |
| Barren | 8.0 | 6.7 | 3.9 | 1.6 | 0.5 | 15.9 |
| Barren | 8.1 | 7.2 | 2.0 | 1.5 | 0.5 | 11.7 |
| Barren | 8.3 | 3.8 | 2.4 | 1.5 | 0.3 | 15.4 |
| Nest | 8.2 | 8.4 | 4.2 | 2.0 | 0.4 | 23.0 |
| Nest | 7.7 | 10.2 | 2.9 | 1.9 | 0.6 | 31.1 |
| Nest | 7.8 | 5.4 | 21.9 | 1.7 | 0.6 | 23.2 |
| Nest | 8.0 | 7.1 | 2.4 | 1.6 | 0.5 | 15.0 |
| Nest | 7.8 | 6.0 | 4.0 | 1.5 | 0.6 | 11.4 |
| Nest | 8.0 | 5.4 | 6.9 | 1.5 | 0.4 | 17.1 |
| Nest | 8.2 | 2.3 | 3.0 | 1.5 | 0.3 | 20.3 |
| Shrub | 8.2 | 5.2 | 4.5 | 1.7 | 0.6 | 25.0 |
| Shrub | 8.2 | 6.0 | 3.8 | 1.7 | 0.8 | 40.2 |
| Shrub | 8.2 | 6.6 | 12.3 | 1.3 | 0.6 | 62.8 |
| Shrub | 8.4 | 4.3 | 1.9 | 1.6 | 0.7 | 13.0 |
| Shrub | 8.3 | 3.4 | 0.9 | 1.4 | 0.6 | 8.4 |
| Shrub | 8.3 | 4.4 | 3.8 | 1.5 | 0.4 | 10.7 |
| Shrub | 8.1 | 4.0 | 5.7 | 1.7 | 0.7 | 22.2 |
| Shrub&Nest | 8.0 | 7.6 | 6.9 | 1.4 | 0.6 | 79.9 |
| Shrub&Nest | 7.7 | 9.5 | 5.3 | 1.5 | 0.8 | 29.4 |
| Shrub&Nest | 7.7 | 11.6 | 42.0 | 1.5 | 0.7 | 76.3 |
| Shrub&Nest | 7.7 | 8.5 | 11.0 | 1.6 | 0.9 | 54.0 |
| Shrub&Nest | 7.8 | 9.6 | 29.8 | 1.4 | 0.9 | 29.0 |
| Shrub&Nest | 7.7 | 14.3 | 105.2 | 1.5 | 0.8 | 66.9 |
| Shrub&Nest | 7.9 | 7.0 | 13.8 | 1.4 | 1.0 | 43.2 |
| Chi2 | 16.5 | 13.9 | 13.1 | 4.7 | 13.3 | 11.5 |







Table A2. P-values of the Dunn Test between patch types on the soil characteristics variables. Bold
numbers are significant (<0.05)

| Comparisons | Water | pH | NO$_3^-$ | NH$_4^+$ | P | OM |
|---|---|---|---|---|---|---|
| **Barren - Nest** | 0.218 | 0.103 | 0.084 | 0.279 | 0.385 | 0.500 |
| **Barren - Shrub** | 0.448 | 0.119 | 0.194 | 0.190 | 0.354 | 0.067 |
| **Nest - Shrub** | 0.181 | **0.007** | 0.301 | 0.072 | 0.468 | 0.067 |
| **Barren - Shrub&Nest** | 0.086 | **0.004** | **0.0003** | **0.004** | **0.001** | **0.001** |
| **Nest - Shrub&Nest** | **0.016** | 0.079 | 0.018 | **0.017** | **0.004** | **0.001** |
| **Shrub - Shrub&Nest** | 0.108 | **0.000** | **0.004** | **0.000** | **0.005** | **0.050** |




Table A3. Results of the pairwise adonis test between patch types done on the NMDS data. Bold
numbers are significant (<0.05).

| Comparison | R2 | P value |
|---|---|---|
| **Control vs Nest** | 0.38473901 | **0.012** |
| **Control vs Shrub** | 0.25759869 | **0.006** |
| **Control vs Shrub&Nest** | 0.21665172 | **0.048** |
| **Nest vs Shrub** | 0.08725184 | 1.000 |
| **Nest vs Shrub&Nest** | 0.21988027 | 0.054 |
| **Shrub vs Shrub&Nest** | 0.08914105 | 1.000 |






Table A4. Number of reads before and after the trimming stage, and during the dada2 stage.

| Sample | Patch Type | Number of reads | | | | |
|---|---|---|---|---|---|---|
| | | Raw | trimmed | filtered | denoised | non-chimeric |
| **Samples_AD1** | Barren | 42089 | 41265 | 36421 | 33675 | 33141 |
| **Samples_AD2** | Barren | 28759 | 28008 | 24434 | 21984 | 21507 |
| **Samples_AD3** | Barren | 30166 | 29410 | 25782 | 23285 | 22830 |
| **Samples_AD4** | Barren | 27024 | 26664 | 23906 | 21545 | 21171 |
| **Samples_AD5** | Barren | 48612 | 47548 | 41813 | 38854 | 38352 |
| **Samples_AD6** | Barren | 23816 | 23120 | 20084 | 18008 | 17857 |
| **Samples_AD7** | Barren | 21806 | 19454 | 16803 | 15532 | 15482 |
| **Samples_AD8** | Nest | 22559 | 20965 | 18485 | 17118 | 17118 |
| **Samples_AD9** | Nest | 28231 | 26041 | 22688 | 21213 | 21088 |
| **Samples_AD10** | Nest | 24428 | 22266 | 19719 | 18340 | 18161 |
| **Samples_AD11** | Nest | 39081 | 37713 | 33573 | 31772 | 31124 |
| **Samples_AD12** | Nest | 18426 | 17446 | 15756 | 14567 | 14494 |
| **Samples_AD13** | Nest | 22881 | 13779 | 10573 | 9234 | 9151 |
| **Samples_AD14** | Nest | 47080 | 44925 | 39700 | 37254 | 36423 |
| **Samples_AD15** | Shrub | 51183 | 48988 | 43764 | 41558 | 40506 |
| **Samples_AD16** | Shrub | 51519 | 37941 | 30791 | 28403 | 27721 |
| **Samples_AD17** | Shrub | 35494 | 33858 | 29858 | 27875 | 27349 |
| **Samples_AD18** | Shrub | 29615 | 27956 | 24841 | 22947 | 22847 |
| **Samples_AD19** | Shrub | 39011 | 37117 | 32622 | 30293 | 29544 |
| **Samples_AD20** | Shrub | 50894 | 38156 | 30901 | 28515 | 28169 |
| **Samples_AD21** | Shrub | 35365 | 32529 | 28933 | 27200 | 27033 |
| **Samples_AD22** | Shrub | 41660 | 27359 | 21466 | 19924 | 19629 |
| **Samples_AD23** | Shrub&Nest | 37107 | 35185 | 31099 | 28722 | 28201 |
| **Samples_AD24** | Shrub&Nest | 55386 | 34724 | 27058 | 24657 | 24136 |
| **Samples_AD25** | Shrub&Nest | 58632 | 42065 | 34139 | 31435 | 30693 |
| **Samples_AD26** | Shrub&Nest | 67273 | 47135 | 37618 | 33503 | 33089 |
| **Samples_AD27** | Shrub&Nest | 35493 | 31891 | 27756 | 26086 | 25915 |
| **Samples_AD28** | Shrub&Nest | 34645 | 29939 | 26141 | 24533 | 24297 |
| **Samples_AD29** | Shrub&Nest | 76888 | 53655 | 42659 | 38753 | 38044 |






Table A5. Relative abundance (%) of the taxonomic community per patch type.

| Phylum | Patch Type | Relative Abundance |
|--------|-----------|--------------------|
| Acidobacteria | Control | 7.02 |
| Acidobacteria | Nest | 2.33 |
| Acidobacteria | Shrub | 5.10 |
| Acidobacteria | Shrub&Nest | 4.52 |
| Actinobacteria | Control | 34.72 |
| Actinobacteria | Nest | 9.79 |
| Actinobacteria | Shrub | 9.13 |
| Actinobacteria | Shrub&Nest | 16.83 |
| Bacteroidetes | Control | 7.41 |
| Bacteroidetes | Nest | 3.86 |
| Bacteroidetes | Shrub | 9.24 |
| Bacteroidetes | Shrub&Nest | 12.42 |
| Chloroflexi | Control | 8.15 |
| Chloroflexi | Nest | 1.01 |
| Chloroflexi | Shrub | 1.75 |
| Chloroflexi | Shrub&Nest | 2.24 |
| Cyanobacteria | Control | 1.59 |
| Cyanobacteria | Shrub | 1.48 |
| Cyanobacteria | Shrub&Nest | 1.95 |
| Deinococcus-Thermus | Control | 2.77 |
| Deinococcus-Thermus | Nest | 30.19 |
| Deinococcus-Thermus | Shrub | 20.85 |
| Deinococcus-Thermus | Shrub&Nest | 8.69 |
| Firmicutes | Control | 1.20 |
| Firmicutes | Nest | 4.89 |
| Firmicutes | Shrub | 6.93 |
| Firmicutes | Shrub&Nest | 9.12 |
| Gemmatimonadetes | Control | 4.93 |
| Gemmatimonadetes | Nest | 1.13 |
| Gemmatimonadetes | Shrub | 2.40 |
| Gemmatimonadetes | Shrub&Nest | 2.78 |
| Planctomycetes | Control | 1.29 |
| Planctomycetes | Nest | 0.55 |
| Planctomycetes | Shrub | 1.39 |
| Planctomycetes | Shrub&Nest | 1.20 |
| Proteobacteria | Control | 27.67 |
| Proteobacteria | Nest | 45.32 |
| Proteobacteria | Shrub | 40.44 |
| Proteobacteria | Shrub&Nest | 38.77 |






Table A6. P-values of the Dunn tests between patch types on the relative abundance of the five most
abundant phyla. Bold numbers are significant (<0.05).

| Comparisons | Actinobacteria | Bacteroidetes | Deinococcus-Thermus | Firmicutes | Proteobacteria |
|---|---|---|---|---|---|
| **Barren - Nest** | **0.0004** | **0.0129** | **0.0003** | 0.3768 | **0.0394** |
| **Barren - Shrub** | **0.0004** | 0.4774 | **0.0009** | 0.0718 | **0.0120** |
| **Nest - Shrub** | 0.4661 | **0.0124** | 0.3352 | 0.1274 | 0.3294 |
| **Barren - Shrub&Nest** | **0.0991** | **0.0836** | **0.0320** | **0.0129** | **0.0042** |
| **Nest - Shrub&Nest** | **0.0207** | **0.0002** | 0.0583 | **0.0278** | 0.1897 |
| **Shrub - Shrub&Nest** | **0.0216** | 0.0690 | 0.1160 | **0.200**8 | 0.3206 |



Table A7. List of the genes used for function prediction ordered by groups and subgroups.

| Group | Metabolic Trait | KEGG ID | Function |
|---|---|---|---|
| DNA conservation | Putative DNA-binding protein | K02524 | K10; DNA binding protein (fs(1)K10, female sterile(1)K10) |
| | Putative DNA-binding protein | K03111 | ssb; single-strand DNA-binding protein |
| | Putative DNA-binding protein | K03530 | hupB; DNA-binding protein HU-beta |
| | Putative DNA-binding protein | K03622 | ssh10b; archaea-specific DNA-binding protein |
| | Putative DNA-binding protein | K03746 | hns; DNA-binding protein H-NS |
| | Putative DNA-binding protein | K04047 | dps; starvation-inducible DNA-binding protein |
| | Putative DNA-binding protein | K04494 | CHD8, HELSNF1; chromodomain helicase DNA binding protein 8 [EC:3.6.4.12] |
| | Putative DNA-binding protein | K04680 | ID1; DNA-binding protein inhibitor ID1 |
| | Putative DNA-binding protein | K05516 | cbpA; curved DNA-binding protein |
| | Putative DNA-binding protein | K05732 | ARHGAP35, GRLF1; glucocorticoid receptor DNA-binding factor 1 |
| | Putative DNA-binding protein | K05787 | hupA; DNA-binding protein HU-alpha |
| | Putative DNA-binding protein | K09061 | GCF, C2orf3; GC-rich sequence DNA-binding factor |
| | Putative DNA-binding protein | K09423 | BAA; Myb-like DNA-binding protein BAA |
| | Putative DNA-binding protein | K09424 | REB1; Myb-like DNA-binding protein REB1 |
| | Putative DNA-binding protein | K09425 | K09425; Myb-like DNA-binding protein FlbD |
| | Putative DNA-binding protein | K09426 | RAP1; Myb-like DNA-binding protein RAP1 |
| | Putative DNA-binding protein | K10140 | DDB2; DNA damage-binding protein 2 |
| | Putative DNA-binding protein | K10610 | DDB1; DNA damage-binding protein 1 |
| | Putative DNA-binding protein | K10728 | TOPBP1; topoisomerase (DNA) II binding protein 1 |



| | Putative DNA-binding protein | K10748 | tus, tau; DNA replication terminus site-binding protein |
|---|---|---|---|
| | Histone-like protein | K10752 | RBBP4, HAT2, CAF1, MIA6; histone-binding protein RBBP4 |
| | Putative DNA-binding protein | K10979 | ku; DNA end-binding protein Ku |
| | Putative DNA-binding protein | K11367 | CHD1; chromodomain-helicase-DNA-binding protein 1 [EC:3.6.4.12] |
| | Histone-like protein | K11495 | CENPA; histone H3-like centromeric protein A |
| | Putative DNA-binding protein | K11574 | CBF2, CBF3A, CTF14; centromere DNA-binding protein complex CBF3 subunit A |
| | Putative DNA-binding protein | K11575 | CEP3, CBF3B; centromere DNA-binding protein complex CBF3 subunit B |
| | Putative DNA-binding protein | K11576 | CTF13, CBF3C; centromere DNA-binding protein complex CBF3 subunit C |
| | Putative DNA-binding protein | K11642 | CHD3, MI2A; chromodomain-helicase-DNA-binding protein 3 [EC:3.6.4.12] |
| | Putative DNA-binding protein | K11643 | CHD4, MI2B; chromodomain-helicase-DNA-binding protein 4 [EC:3.6.4.12] |
| | Histone-like protein | K11659 | RBBP7; histone-binding protein RBBP7 |
| | Putative DNA-binding protein | K11685 | stpA; DNA-binding protein StpA |
| | Putative DNA-binding protein | K12965 | ZBP1, DAI; Z-DNA binding protein 1 |
| | Putative DNA-binding protein | K13102 | KIN; DNA/RNA-binding protein KIN17 |
| | Putative DNA-binding protein | K13211 | GCFC; GC-rich sequence DNA-binding factor |
| | Putative DNA-binding protein | K14435 | CHD5; chromodomain-helicase-DNA-binding protein 5 [EC:3.6.4.12] |
| | Putative DNA-binding protein | K14436 | CHD6; chromodomain-helicase-DNA-binding protein 6 [EC:3.6.4.12] |
| | Putative DNA-binding protein | K14437 | CHD7; chromodomain-helicase-DNA-binding protein 7 [EC:3.6.4.12] |
| | Putative DNA-binding protein | K14438 | CHD9; chromodomain-helicase-DNA-binding protein 9 [EC:3.6.4.12] |
| | Putative DNA-binding protein | K14507 | ORCA2_3; AP2-domain DNA-binding protein ORCA2/3 |
| | Histone-like protein | K15719 | NCOAT, MGEA5; protein O-GlcNAcase / histone acetyltransferase [EC:3.2.1.169 2.3.1.48] |
| | Putative DNA-binding protein | K16640 | ssh7; DNA-binding protein 7 [EC:3.1.27.-] |
| | Putative DNA-binding protein | K17693 | ID2; DNA-binding protein inhibitor ID2 |
| | Putative DNA-binding protein | K17694 | ID3; DNA-binding protein inhibitor ID3 |
| | Putative DNA-binding protein | K17695 | ID4; DNA-binding protein inhibitor ID4 |
| | Putative DNA-binding protein | K17696 | EMC; DNA-binding protein inhibitor ID, other |
| | Histone-like protein | K18710 | SLBP; histone RNA hairpin-binding protein |



| | Putative DNA-binding protein | K18946 | gp32, ssb; single-stranded DNA-binding protein |
|---|---|---|---|
| | Putative DNA-binding protein | K19442 | ICP8, DBP, UL29; Simplexvirus major DNA-binding protein |
| | Histone-like protein | K19799 | RPH1; DNA damage-responsive transcriptional repressor / [histone H3]-trimethyl-L-lysine36 demethylase [EC:1.14.11.69] |
| | Putative DNA-binding protein | K20091 | CHD2; chromodomain-helicase-DNA-binding protein 2 [EC:3.6.4.12] |
| | Putative DNA-binding protein | K20092 | CHD1L; chromodomain-helicase-DNA-binding protein 1-like [EC:3.6.4.12] |
| | Putative DNA-binding protein | K22592 | AHDC1; AT-hook DNA-binding motif-containing protein 1 |
| | Putative DNA-binding protein | K23225 | SATB1; DNA-binding protein SATB1 |
| | Putative DNA-binding protein | K23226 | SATB2; DNA-binding protein SATB2 |
| | Putative DNA-binding protein | K23600 | TARDBP, TDP43; TAR DNA-binding protein 43 |
| DNA repair | DNA polymerase PolA (COG0258) | K02320 | POLA1; DNA polymerase alpha subunit A [EC:2.7.7.7] |
| | DNA polymerase PolA (COG0258) | K02321 | POLA2; DNA polymerase alpha subunit B |
| | DNA polymerase PolA (COG0258) | K02335 | polA; DNA polymerase I [EC:2.7.7.7] |
| | DNA polymerase IV | K02346 | dinB; DNA polymerase IV [EC:2.7.7.7] |
| | Exodeoxyribonuclease VII | K03601 | xseA; exodeoxyribonuclease VII large subunit [EC:3.1.11.6] |
| | Exodeoxyribonuclease VII | K03602 | xseB; exodeoxyribonuclease VII small subunit [EC:3.1.11.6] |
| | DNA polymerase IV | K04479 | dbh; DNA polymerase IV (archaeal DinB-like DNA polymerase) [EC:2.7.7.7] |
| | Exodeoxyribonuclease VII | K10906 | recE; exodeoxyribonuclease VIII [EC:3.1.11.-] |
| | DNA polymerase IV | K10981 | POL4; DNA polymerase IV [EC:2.7.7.7] |
| | DNA polymerase IV | K16250 | NRPD1; DNA-directed RNA polymerase IV subunit 1 [EC:2.7.7.6] |
| | DNA polymerase IV | K16252 | NRPD2, NRPE2; DNA-directed RNA polymerase IV and V subunit 2 [EC:2.7.7.6] |
| | DNA polymerase IV | K16253 | NRPD7, NRPE7; DNA-directed RNA polymerase IV and V subunit 7 |
| Lithotrophy | NiFe hydrogenase | K00437 | hydB; [NiFe] hydrogenase large subunit [EC:1.12.2.1] |
| | NiFe hydrogenase | K02587 | nifE; nitrogenase molybdenum-cofactor synthesis protein NifE |
| | CO-dehydrogenase CoxM & CoxS | K03518 | coxS; aerobic carbon-monoxide dehydrogenase small subunit [EC:1.2.5.3] |
| | CO-dehydrogenase CoxM & CoxS | K03519 | coxM, cutM; aerobic carbon-monoxide dehydrogenase medium subunit [EC:1.2.5.3] |
| | CO-dehydrogenase large subunit (coxL) Form I | K03520 | coxL, cutL; aerobic carbon-monoxide dehydrogenase large subunit [EC:1.2.5.3] |



| | | | |
|---|---|---|---|
| | NiFe hydrogenase | K05586 | hoxE; bidirectional [NiFe] hydrogenase diaphorase subunit [EC:7.1.1.2] |
| | NiFe hydrogenase | K05587 | hoxF; bidirectional [NiFe] hydrogenase diaphorase subunit [EC:7.1.1.2] |
| | NiFe hydrogenase | K05588 | hoxU; bidirectional [NiFe] hydrogenase diaphorase subunit [EC:7.1.1.2] |
| | SOX sulfur-oxidation system | K17218 | sqr; sulfide:quinone oxidoreductase [EC:1.8.5.4] |
| | SOX sulfur-oxidation system | K17222 | soxA; L-cysteine S-thiosulfotransferase [EC:2.8.5.2] |
| | SOX sulfur-oxidation system | K17223 | soxX; L-cysteine S-thiosulfotransferase [EC:2.8.5.2] |
| | SOX sulfur-oxidation system | K17224 | soxB; S-sulfosulfanyl-L-cysteine sulfohydrolase [EC:3.1.6.20] |
| | SOX sulfur-oxidation system | K17225 | soxC; sulfane dehydrogenase subunit SoxC |
| | SOX sulfur-oxidation system | K17226 | soxY; sulfur-oxidizing protein SoxY |
| | SOX sulfur-oxidation system | K17227 | soxZ; sulfur-oxidizing protein SoxZ |
| | NiFe hydrogenase | K18005 | hoxF; [NiFe] hydrogenase diaphorase moiety large subunit [EC:1.12.1.2] |
| | NiFe hydrogenase | K18006 | hoxU; [NiFe] hydrogenase diaphorase moiety small subunit [EC:1.12.1.2] |
| | NiFe hydrogenase | K18008 | hydA; [NiFe] hydrogenase small subunit [EC:1.12.2.1] |
| | Propane monooxygenase (soluble) | K18223 | prmA; propane 2-monooxygenase large subunit [EC:1.14.13.227] |
| | Propane monooxygenase (soluble) | K18224 | prmC; propane 2-monooxygenase small subunit [EC:1.14.13.227] |
| | Propane monooxygenase (soluble) | K18225 | prmB; propane monooxygenase reductase component [EC:1.18.1.-] |
| | Propane monooxygenase (soluble) | K18226 | prmD; propane monooxygenase coupling protein |
| | SOX sulfur-oxidation system | K22622 | soxD; S-disulfanyl-L-cysteine oxidoreductase SoxD [EC:1.8.2.6] |
| | SOX sulfur-oxidation system | K24007 | soxD; cytochrome aa3-type oxidase subunit SoxD |
| | SOX sulfur-oxidation system | K24008 | soxC; cytochrome aa3-type oxidase subunit III |
| | SOX sulfur-oxidation system | K24009 | soxB; cytochrome aa3-type oxidase subunit I [EC:7.1.1.4] |
| | SOX sulfur-oxidation system | K24010 | soxA; cytochrome aa3-type oxidase subunit II [EC:7.1.1.4] |
| | SOX sulfur-oxidation system | K24011 | soxM; cytochrome aa3-type oxidase subunit I/III [EC:7.1.1.4] |
| Organotrophy | ABC sugar transporters | K02025 | ABC.MS.P; multiple sugar transport system permease protein |
| | ABC sugar transporters | K02026 | ABC.MS.P1; multiple sugar transport system permease protein |
| | ABC sugar transporters | K02027 | ABC.MS.S; multiple sugar transport system substrate-binding protein |
| | ABC sugar transporters | K02056 | ABC.SS.A; simple sugar transport system ATP-binding protein [EC:7.5.2.-] |
| | ABC sugar transporters | K02057 | ABC.SS.P; simple sugar transport system permease protein |





| | | | |
|---|---|---|---|
| ABC sugar transporters | K02058 | ABC.SS.S; simple sugar transport system substrate-binding protein |
| PTS sugar importers | K02777 | crr; sugar PTS system EIIA component [EC:2.7.1.-] |
| Amino acid transporter | K03293 | TC.AAT; amino acid transporter, AAT family |
| Peptide transporter | K03305 | TC.POT; proton-dependent oligopeptide transporter, POT family |
| Amino acid transporter | K03311 | TC.LIVCS; branched-chain amino acid:cation transporter, LIVCS family |
| Carboxylate transporters | K03326 | TC.DCUC, dcuC, dcuD; C4-dicarboxylate transporter, DcuC family |
| Amino acid transporter | K03450 | SLC7A; solute carrier family 7 (L-type amino acid transporter), other |
| Glycosyl hydrolases | K04844 | ycjT; hypothetical glycosyl hydrolase [EC:3.2.1.-] |
| Amino acid transporter | K05048 | SLC6A15S; solute carrier family 6 (neurotransmitter transporter, amino acid/orphan) member 15/16/17/18/20 |
| Amino acid transporter | K05615 | SLC1A4, SATT; solute carrier family 1 (neutral amino acid transporter), member 4 |
| Amino acid transporter | K05616 | SLC1A5; solute carrier family 1 (neutral amino acid transporter), member 5 |
| Amino acid transporter | K07084 | yuiF; putative amino acid transporter |
| Carboxylate transporters | K07791 | dcuA; anaerobic C4-dicarboxylate transporter DcuA |
| Carboxylate transporters | K07792 | dcuB; anaerobic C4-dicarboxylate transporter DcuB |
| ABC sugar transporters | K10546 | ABC.GGU.S, chvE; putative multiple sugar transport system substrate-binding protein |
| ABC sugar transporters | K10547 | ABC.GGU.P, gguB; putative multiple sugar transport system permease protein |
| ABC sugar transporters | K10548 | ABC.GGU.A, gguA; putative multiple sugar transport system ATP-binding protein [EC:7.5.2.-] |
| Carboxylate transporters | K11689 | dctQ; C4-dicarboxylate transporter, DctQ subunit |
| Carboxylate transporters | K11690 | dctM; C4-dicarboxylate transporter, DctM subunit |
| Amino acid transporter | K13576 | SLC38A3, SNAT3; solute carrier family 38 (sodium-coupled neutral amino acid transporter), member 3 |
| Carboxylate transporters | K13577 | SLC25A10, DIC; solute carrier family 25 (mitochondrial dicarboxylate transporter), member 10 |
| Amino acid transporter | K13780 | SLC7A5, LAT1; solute carrier family 7 (L-type amino acid transporter), member 5 |
| Amino acid transporter | K13781 | SLC7A8, LAT2; solute carrier family 7 (L-type amino acid transporter), member 8 |
| Amino acid transporter | K13782 | SLC7A10, ASC1; solute carrier family 7 (L-type amino acid transporter), member 10 |



| | Amino acid transporter | K13863 | SLC7A1, ATRC1; solute carrier family 7 (cationic amino acid transporter), member 1 |
|---|---|---|---|
| | Amino acid transporter | K13864 | SLC7A2, ATRC2; solute carrier family 7 (cationic amino acid transporter), member 2 |
| | Amino acid transporter | K13865 | SLC7A3, ATRC3; solute carrier family 7 (cationic amino acid transporter), member 3 |
| | Amino acid transporter | K13866 | SLC7A4; solute carrier family 7 (cationic amino acid transporter), member 4 |
| | Amino acid transporter | K13867 | SLC7A7; solute carrier family 7 (L-type amino acid transporter), member 7 |
| | Amino acid transporter | K13868 | SLC7A9, BAT1; solute carrier family 7 (L-type amino acid transporter), member 9 |
| | Amino acid transporter | K13869 | SLC7A11; solute carrier family 7 (L-type amino acid transporter), member 11 |
| | Amino acid transporter | K13870 | SLC7A13, AGT1; solute carrier family 7 (L-type amino acid transporter), member 13 |
| | Amino acid transporter | K13871 | SLC7A14; solute carrier family 7 (cationic amino acid transporter), member 14 |
| | Amino acid transporter | K13872 | SLC7A6; solute carrier family 7 (L-type amino acid transporter), member 6 |
| | Peptide transporter | K14206 | SLC15A1, PEPT1; solute carrier family 15 (oligopeptide transporter), member 1 |
| | Amino acid transporter | K14207 | SLC38A2, SNAT2; solute carrier family 38 (sodium-coupled neutral amino acid transporter), member 2 |
| | Amino acid transporter | K14209 | SLC36A, PAT; solute carrier family 36 (proton-coupled amino acid transporter) |
| | Amino acid transporter | K14210 | SLC3A1, RBAT; solute carrier family 3 (neutral and basic amino acid transporter), member 1 |
| | Carboxylate transporters | K14388 | SLC5A8_12, SMCT; solute carrier family 5 (sodium-coupled monocarboxylate transporter), member 8/12 |
| | Carboxylate transporters | K14445 | SLC13A2_3_5; solute carrier family 13 (sodium-dependent dicarboxylate transporter), member 2/3/5 |
| | Peptide transporter | K14637 | SLC15A2, PEPT2; solute carrier family 15 (oligopeptide transporter), member 2 |
| | Peptide transporter | K14638 | SLC15A3_4, PHT; solute carrier family 15 (peptide/histidine transporter), member 3/4 |
| | Amino acid transporter | K14990 | SLC38A1, SNAT1, GLNT; solute carrier family 38 (sodium-coupled neutral amino acid transporter), member 1 |





| | | | |
|---|---|---|---|
| | Amino acid transporter | K14991 | SLC38A4, SNAT4; solute carrier family 38 (sodium-coupled neutral amino acid transporter), member 4 |
| | Amino acid transporter | K14992 | SLC38A5, SNAT5; solute carrier family 38 (sodium-coupled neutral amino acid transporter), member 5 |
| | Amino acid transporter | K14993 | SLC38A6, SNAT6; solute carrier family 38 (sodium-coupled neutral amino acid transporter), member 6 |
| | Amino acid transporter | K14994 | SLC38A7_8; solute carrier family 38 (sodium-coupled neutral amino acid transporter), member 7/8 |
| | Amino acid transporter | K14995 | SLC38A9; solute carrier family 38 (sodium-coupled neutral amino acid transporter), member 9 |
| | Amino acid transporter | K14996 | SLC38A10; solute carrier family 38 (sodium-coupled neutral amino acid transporter), member 10 |
| | Amino acid transporter | K14997 | SLC38A11; solute carrier family 38 (sodium-coupled neutral amino acid transporter), member 11 |
| | Amino acid transporter | K15015 | SLC32A, VGAT; solute carrier family 32 (vesicular inhibitory amino acid transporter) |
| | Carboxylate transporters | K15110 | SLC25A21, ODC; solute carrier family 25 (mitochondrial 2-oxodicarboxylate transporter), member 21 |
| | Amino acid transporter | K16261 | YAT; yeast amino acid transporter |
| | Amino acid transporter | K16263 | yjeH; amino acid efflux transporter |
| | Peptide transporter | K17938 | sbmA, bacA; peptide/bleomycin uptake transporter |
| Phototrophy | RuBisCO | K01601 | rbcL; ribulose-bisphosphate carboxylase large chain [EC:4.1.1.39] |
| | Chlorophyll synthesis | K01669 | phrB; deoxyribodipyrimidine photo-lyase [EC:4.1.99.3] |
| | Chlorophyll synthesis | K02689 | psaA; photosystem I P700 chlorophyll a apoprotein A1 |
| | Chlorophyll synthesis | K02690 | psaB; photosystem I P700 chlorophyll a apoprotein A2 |
| | Chlorophyll synthesis | K02691 | psaC; photosystem I subunit VII |
| | Chlorophyll synthesis | K02692 | psaD; photosystem I subunit II |
| | Chlorophyll synthesis | K02693 | psaE; photosystem I subunit IV |
| | Chlorophyll synthesis | K02694 | psaF; photosystem I subunit III |
| | Chlorophyll synthesis | K02695 | psaH; photosystem I subunit VI |
| | Chlorophyll synthesis | K02696 | psaI; photosystem I subunit VIII |
| | Chlorophyll synthesis | K02697 | psaJ; photosystem I subunit IX |
| | Chlorophyll synthesis | K02698 | psaK; photosystem I subunit X |
| | Chlorophyll synthesis | K02699 | psaL; photosystem I subunit XI |
| | Chlorophyll synthesis | K02700 | psaM; photosystem I subunit XII |
| | Chlorophyll synthesis | K02701 | psaN; photosystem I subunit PsaN |
| | Chlorophyll synthesis | K02702 | psaX; photosystem I 4.8kDa protein |
| | Chlorophyll synthesis | K02703 | psbA; photosystem II P680 reaction center D1 protein [EC:1.10.3.9] |
| | Chlorophyll synthesis | K02704 | psbB; photosystem II CP47 chlorophyll apoprotein |





| | Chlorophyll synthesis | K02705 | psbC; photosystem II CP43 chlorophyll apoprotein |
|---|---|---|---|
| | Chlorophyll synthesis | K02706 | psbD; photosystem II P680 reaction center D2 protein [EC:1.10.3.9] |
| | Chlorophyll synthesis | K02707 | psbE; photosystem II cytochrome b559 subunit alpha |
| | Chlorophyll synthesis | K02708 | psbF; photosystem II cytochrome b559 subunit beta |
| | Chlorophyll synthesis | K02709 | psbH; photosystem II PsbH protein |
| | Chlorophyll synthesis | K02710 | psbI; photosystem II PsbI protein |
| | Chlorophyll synthesis | K02711 | psbJ; photosystem II PsbJ protein |
| | Chlorophyll synthesis | K02712 | psbK; photosystem II PsbK protein |
| | Chlorophyll synthesis | K02713 | psbL; photosystem II PsbL protein |
| | Chlorophyll synthesis | K02714 | psbM; photosystem II PsbM protein |
| | Chlorophyll synthesis | K02716 | psbO; photosystem II oxygen-evolving enhancer protein 1 |
| | Chlorophyll synthesis | K02717 | psbP; photosystem II oxygen-evolving enhancer protein 2 |
| | Chlorophyll synthesis | K02718 | psbT; photosystem II PsbT protein |
| | Chlorophyll synthesis | K02719 | psbU; photosystem II PsbU protein |
| | Chlorophyll synthesis | K02720 | psbV; photosystem II cytochrome c550 |
| | Chlorophyll synthesis | K02721 | psbW; photosystem II PsbW protein |
| | Chlorophyll synthesis | K02722 | psbX; photosystem II PsbX protein |
| | Chlorophyll synthesis | K02723 | psbY; photosystem II PsbY protein |
| | Chlorophyll synthesis | K02724 | psbZ; photosystem II PsbZ protein |
| | Chlorophyll synthesis | K03157 | LTB, TNFC; lymphotoxin beta (TNF superfamily, member 3) |
| | Chlorophyll synthesis | K03159 | TNFRSF3, LTBR; lymphotoxin beta receptor TNFR superfamily member 3 |
| | Chlorophyll synthesis | K03541 | psbR; photosystem II 10kDa protein |
| | Chlorophyll synthesis | K03542 | psbS; photosystem II 22kDa protein |
| | Chlorophyll synthesis | K03716 | splB; spore photoproduct lyase [EC:4.1.99.14] |
| | Chlorophyll synthesis | K05468 | LTA, TNFB; lymphotoxin alpha (TNF superfamily, member 1) |
| | Chlorophyll synthesis | K06315 | splA; transcriptional regulator of the spore photoproduct lyase operon |
| | Chlorophyll synthesis | K06876 | K06876; deoxyribodipyrimidine photolyase-related protein |
| | Chlorophyll synthesis | K08901 | psbQ; photosystem II oxygen-evolving enhancer protein 3 |
| | Chlorophyll synthesis | K08902 | psb27; photosystem II Psb27 protein |
| | Chlorophyll synthesis | K08903 | psb28; photosystem II 13kDa protein |
| | Chlorophyll synthesis | K08904 | psb28-2; photosystem II Psb28-2 protein |
| | Chlorophyll synthesis | K08905 | psaG; photosystem I subunit V |
| | Chlorophyll synthesis | K08928 | pufL; photosynthetic reaction center L subunit |
| | Chlorophyll synthesis | K08929 | pufM; photosynthetic reaction center M subunit |
| | Chlorophyll synthesis | K08940 | pscA; photosystem P840 reaction center large subunit |
| | Chlorophyll synthesis | K08941 | pscB; photosystem P840 reaction center iron-sulfur protein |



| | | | |
|---|---|---|---|
| | Chlorophyll synthesis | K08942 | pscC; photosystem P840 reaction center cytochrome c551 |
| | Chlorophyll synthesis | K08943 | pscD; photosystem P840 reaction center protein PscD |
| | Chlorophyll synthesis | K11524 | pixI; positive phototaxis protein PixI |
| | Chlorophyll synthesis | K13991 | puhA; photosynthetic reaction center H subunit |
| | Chlorophyll synthesis | K13992 | pufC; photosynthetic reaction center cytochrome c subunit |
| | Chlorophyll synthesis | K13994 | pufX; photosynthetic reaction center PufX protein |
| | Chlorophyll synthesis | K14332 | psaO; photosystem I subunit PsaO |
| | Chlorophyll synthesis | K19016 | IMPG1, SPACR; interphotoreceptor matrix proteoglycan 1 |
| | Chlorophyll synthesis | K19017 | IMPG2, SPACRCAN; interphotoreceptor matrix proteoglycan 2 |
| | Chlorophyll synthesis | K20715 | PHOT; phototropin [EC:2.7.11.1] |
| | Chlorophyll synthesis | K22464 | FAP; fatty acid photodecarboxylase [EC:4.1.1.106] |
| | Chlorophyll synthesis | K22619 | Aequorin; calcium-regulated photoprotein [EC:1.13.12.24] |
| | Chlorophyll synthesis | K24165 | PCARE; photoreceptor cilium actin regulator |
| ROS-damage prevention | Cytochrome C oxidase | K00404 | ccoN; cytochrome c oxidase cbb3-type subunit I [EC:7.1.1.9] |
| | Cytochrome C oxidase | K00405 | ccoO; cytochrome c oxidase cbb3-type subunit II |
| | Cytochrome C oxidase | K00406 | ccoP; cytochrome c oxidase cbb3-type subunit III |
| | Cytochrome C oxidase | K00407 | ccoQ; cytochrome c oxidase cbb3-type subunit IV |
| | Cytochrome bd ubiquinol oxidase | K00424 | cydX; cytochrome bd-I ubiquinol oxidase subunit X [EC:7.1.1.7] |
| | Cytochrome C oxidase | K00424 | cydX; cytochrome bd-I ubiquinol oxidase subunit X [EC:7.1.1.7] |
| | Cytochrome bd ubiquinol oxidase | K00425 | cydA; cytochrome bd ubiquinol oxidase subunit I [EC:7.1.1.7] |
| | Cytochrome C oxidase | K00425 | cydA; cytochrome bd ubiquinol oxidase subunit I [EC:7.1.1.7] |
| | Cytochrome bd ubiquinol oxidase | K00426 | cydB; cytochrome bd ubiquinol oxidase subunit II [EC:7.1.1.7] |
| | Cytochrome C oxidase | K00426 | cydB; cytochrome bd ubiquinol oxidase subunit II [EC:7.1.1.7] |
| | Cytochrome C oxidase | K00428 | E1.11.1.5; cytochrome c peroxidase [EC:1.11.1.5] |
| | Cytochrome C oxidase | K02256 | COX1; cytochrome c oxidase subunit 1 [EC:7.1.1.9] |
| | Cytochrome C oxidase | K02258 | COX11, ctaG; cytochrome c oxidase assembly protein subunit 11 |
| | Cytochrome C oxidase | K02259 | COX15, ctaA; cytochrome c oxidase assembly protein subunit 15 |
| | Cytochrome C oxidase | K02260 | COX17; cytochrome c oxidase assembly protein subunit 17 |
| | Cytochrome C oxidase | K02261 | COX2; cytochrome c oxidase subunit 2 |
| | Cytochrome C oxidase | K02262 | COX3; cytochrome c oxidase subunit 3 |





| | Cytochrome C oxidase | K02263 | COX4; cytochrome c oxidase subunit 4 |
| --- | --- | --- | --- |
| | Cytochrome C oxidase | K02264 | COX5A; cytochrome c oxidase subunit 5a |
| | Cytochrome C oxidase | K02265 | COX5B; cytochrome c oxidase subunit 5b |
| | Cytochrome C oxidase | K02266 | COX6A; cytochrome c oxidase subunit 6a |
| | Cytochrome C oxidase | K02267 | COX6B; cytochrome c oxidase subunit 6b |
| | Cytochrome C oxidase | K02268 | COX6C; cytochrome c oxidase subunit 6c |
| | Cytochrome C oxidase | K02269 | COX7; cytochrome c oxidase subunit 7 |
| | Cytochrome C oxidase | K02270 | COX7A; cytochrome c oxidase subunit 7a |
| | Cytochrome C oxidase | K02271 | COX7B; cytochrome c oxidase subunit 7b |
| | Cytochrome C oxidase | K02272 | COX7C; cytochrome c oxidase subunit 7c |
| | Cytochrome C oxidase | K02273 | COX8; cytochrome c oxidase subunit 8 |
| | Cytochrome C oxidase | K02274 | coxA, ctaD; cytochrome c oxidase subunit I [EC:7.1.1.9] |
| | Cytochrome C oxidase | K02275 | coxB, ctaC; cytochrome c oxidase subunit II [EC:7.1.1.9] |
| | Cytochrome C oxidase | K02276 | coxC, ctaE; cytochrome c oxidase subunit III [EC:7.1.1.9] |
| | Cytochrome C oxidase | K02277 | coxD, ctaF; cytochrome c oxidase subunit IV [EC:7.1.1.9] |
| | Cytochrome C oxidase | K02297 | cyoA; cytochrome o ubiquinol oxidase subunit II [EC:7.1.1.3] |
| | Cytochrome C oxidase | K02298 | cyoB; cytochrome o ubiquinol oxidase subunit I [EC:7.1.1.3] |
| | Cytochrome C oxidase | K02299 | cyoC; cytochrome o ubiquinol oxidase subunit III |
| | Cytochrome C oxidase | K02300 | cyoD; cytochrome o ubiquinol oxidase subunit IV |
| | Cytochrome C oxidase | K02826 | qoxA; cytochrome aa3-600 menaquinol oxidase subunit II [EC:7.1.1.5] |
| | Cytochrome C oxidase | K02827 | qoxB; cytochrome aa3-600 menaquinol oxidase subunit I [EC:7.1.1.5] |
| | Cytochrome C oxidase | K02828 | qoxC; cytochrome aa3-600 menaquinol oxidase subunit III [EC:7.1.1.5] |
| | Cytochrome C oxidase | K02829 | qoxD; cytochrome aa3-600 menaquinol oxidase subunit IV [EC:7.1.1.5] |
| | Mn2+ catalase | K07217 | K07217; Mn-containing catalase |
| | Cytochrome C oxidase | K15408 | coxAC; cytochrome c oxidase subunit I+III [EC:7.1.1.9] |
| | Cytochrome C oxidase | K15862 | ccoNO; cytochrome c oxidase cbb3-type subunit I/II [EC:7.1.1.9] |
| | Cytochrome C oxidase | K18173 | COA1; cytochrome c oxidase assembly factor 1 |
| | Cytochrome C oxidase | K18174 | COA2; cytochrome c oxidase assembly factor 2 |



| | | | |
|---|---|---|---|
| | Cytochrome C oxidase | K18175 | CCDC56, COA3; cytochrome c oxidase assembly factor 3, animal type |
| | Cytochrome C oxidase | K18176 | COA3; cytochrome c oxidase assembly factor 3, fungi type |
| | Cytochrome C oxidase | K18177 | COA4; cytochrome c oxidase assembly factor 4 |
| | Cytochrome C oxidase | K18178 | COA5, PET191; cytochrome c oxidase assembly factor 5 |
| | Cytochrome C oxidase | K18179 | COA6; cytochrome c oxidase assembly factor 6 |
| | Cytochrome C oxidase | K18180 | COA7, SELRC1, RESA1; cytochrome c oxidase assembly factor 7 |
| | Cytochrome C oxidase | K18181 | COX14; cytochrome c oxidase assembly factor 14 |
| | Cytochrome C oxidase | K18182 | COX16; cytochrome c oxidase assembly protein subunit 16 |
| | Cytochrome C oxidase | K18183 | COX19; cytochrome c oxidase assembly protein subunit 19 |
| | Cytochrome C oxidase | K18184 | COX20; cytochrome c oxidase assembly protein subunit 20 |
| | Cytochrome C oxidase | K18185 | COX23; cytochrome c oxidase assembly protein subunit 23 |
| | Cytochrome C oxidase | K18189 | TACO1; translational activator of cytochrome c oxidase 1 |
| | Cytochrome bd ubiquinol oxidase | K22501 | appX; cytochrome bd-II ubiquinol oxidase subunit AppX [EC:7.1.1.7] |
| | Cytochrome C oxidase | K22501 | appX; cytochrome bd-II ubiquinol oxidase subunit AppX [EC:7.1.1.7] |
| | Cytochrome C oxidase | K24007 | soxD; cytochrome aa3-type oxidase subunit SoxD |
| | Cytochrome C oxidase | K24008 | soxC; cytochrome aa3-type oxidase subunit III |
| | Cytochrome C oxidase | K24009 | soxB; cytochrome aa3-type oxidase subunit I [EC:7.1.1.4] |
| | Cytochrome C oxidase | K24010 | soxA; cytochrome aa3-type oxidase subunit II [EC:7.1.1.4] |
| | Cytochrome C oxidase | K24011 | soxM; cytochrome aa3-type oxidase subunit I/III [EC:7.1.1.4] |
| Sporulation | Glycogen synthesis | K00693 | GYS; glycogen synthase [EC:2.4.1.11] |
| | Sporulation (Actinobacteria) | K02490 | spo0F; two-component system, response regulator, stage 0 sporulation protein F |
| | Sporulation (Actinobacteria) | K02491 | kinA; two-component system, sporulation sensor kinase A [EC:2.7.13.3] |
| | Glycogen synthesis | K03083 | GSK3B; glycogen synthase kinase 3 beta [EC:2.7.11.26] |
| | Sporulation (Actinobacteria) | K03091 | sigH; RNA polymerase sporulation-specific sigma factor |
| | Sporulation (Actinobacteria) | K04769 | spoVT; AbrB family transcriptional regulator, stage V sporulation protein T |
| | Sporulation (Actinobacteria) | K06283 | spoIIID; putative DeoR family transcriptional regulator, stage III sporulation protein D |
| | Sporulation (Actinobacteria) | K06348 | kapD; sporulation inhibitor KapD |
| | Sporulation (Actinobacteria) | K06359 | rapA, spo0L; response regulator aspartate phosphatase A (stage 0 sporulation protein L) [EC:3.1.-.-] |



| | | | |
|---|---|---|---|
| Sporulation (Actinobacteria) | K06371 | sda; developmental checkpoint coupling sporulation initiation to replication initiation |
| Sporulation (Actinobacteria) | K06375 | spo0B; stage 0 sporulation protein B (sporulation initiation phosphotransferase) [EC:2.7.-.-] |
| Sporulation (Actinobacteria) | K06376 | spo0E; stage 0 sporulation regulatory protein |
| Sporulation (Actinobacteria) | K06377 | spo0M; sporulation-barren protein |
| Sporulation (Actinobacteria) | K06378 | spoIIAA; stage II sporulation protein AA (anti-sigma F factor antagonist) |
| Sporulation (Actinobacteria) | K06379 | spoIIAB; stage II sporulation protein AB (anti-sigma F factor) [EC:2.7.11.1] |
| Sporulation (Actinobacteria) | K06380 | spoIIB; stage II sporulation protein B |
| Sporulation (Actinobacteria) | K06381 | spoIID; stage II sporulation protein D |
| Sporulation (Actinobacteria) | K06382 | spoIIE; stage II sporulation protein E [EC:3.1.3.16] |
| Sporulation (Actinobacteria) | K06383 | spoIIGA; stage II sporulation protein GA (sporulation sigma-E factor processing peptidase) [EC:3.4.23.-] |
| Sporulation (Actinobacteria) | K06384 | spoIIM; stage II sporulation protein M |
| Sporulation (Actinobacteria) | K06385 | spoIIP; stage II sporulation protein P |
| Sporulation (Actinobacteria) | K06386 | spoIIQ; stage II sporulation protein Q |
| Sporulation (Actinobacteria) | K06387 | spoIIR; stage II sporulation protein R |
| Sporulation (Actinobacteria) | K06388 | spoIISA; stage II sporulation protein SA |
| Sporulation (Actinobacteria) | K06389 | spoIISB; stage II sporulation protein SB |
| Sporulation (Actinobacteria) | K06390 | spoIIIAA; stage III sporulation protein AA |
| Sporulation (Actinobacteria) | K06391 | spoIIIAB; stage III sporulation protein AB |
| Sporulation (Actinobacteria) | K06392 | spoIIIAC; stage III sporulation protein AC |
| Sporulation (Actinobacteria) | K06393 | spoIIIAD; stage III sporulation protein AD |
| Sporulation (Actinobacteria) | K06394 | spoIIIAE; stage III sporulation protein AE |
| Sporulation (Actinobacteria) | K06395 | spoIIIAF; stage III sporulation protein AF |
| Sporulation (Actinobacteria) | K06396 | spoIIIAG; stage III sporulation protein AG |
| Sporulation (Actinobacteria) | K06397 | spoIIIAH; stage III sporulation protein AH |
| Sporulation (Actinobacteria) | K06398 | spoIVA; stage IV sporulation protein A |
| Sporulation (Actinobacteria) | K06399 | spoIVB; stage IV sporulation protein B [EC:3.4.21.116] |
| Sporulation (Actinobacteria) | K06401 | spoIVFA; stage IV sporulation protein FA |
| Sporulation (Actinobacteria) | K06402 | spoIVFB; stage IV sporulation protein FB [EC:3.4.24.-] |
| Sporulation (Actinobacteria) | K06403 | spoVAA; stage V sporulation protein AA |
| Sporulation (Actinobacteria) | K06404 | spoVAB; stage V sporulation protein AB |





| | Sporulation (Actinobacteria) | K06405 | spoVAC; stage V sporulation protein AC |
|---|---|---|---|
| | Sporulation (Actinobacteria) | K06406 | spoVAD; stage V sporulation protein AD |
| | Sporulation (Actinobacteria) | K06407 | spoVAE; stage V sporulation protein AE |
| | Sporulation (Actinobacteria) | K06408 | spoVAF; stage V sporulation protein AF |
| | Sporulation (Actinobacteria) | K06409 | spoVB; stage V sporulation protein B |
| | Sporulation (Actinobacteria) | K06412 | spoVG; stage V sporulation protein G |
| | Sporulation (Actinobacteria) | K06413 | spoVK; stage V sporulation protein K |
| | Sporulation (Actinobacteria) | K06414 | spoVM; stage V sporulation protein M |
| | Sporulation (Actinobacteria) | K06415 | spoVR; stage V sporulation protein R |
| | Sporulation (Actinobacteria) | K06416 | spoVS; stage V sporulation protein S |
| | Sporulation (Actinobacteria) | K06417 | spoVID; stage VI sporulation protein D |
| | Sporulation (Actinobacteria) | K06437 | yknT; sigma-E barrenled sporulation protein |
| | Sporulation (Actinobacteria) | K06438 | yqfD; similar to stage IV sporulation protein |
| | Sporulation (Actinobacteria) | K07697 | kinB; two-component system, sporulation sensor kinase B [EC:2.7.13.3] |
| | Sporulation (Actinobacteria) | K07698 | kinC; two-component system, sporulation sensor kinase C [EC:2.7.13.3] |
| | Sporulation (Actinobacteria) | K07699 | spo0A; two-component system, response regulator, stage 0 sporulation protein A |
| | Sporulation (Actinobacteria) | K08293 | SMK1; sporulation-specific mitogen-activated protein kinase SMK1 [EC:2.7.11.24] |
| | Sporulation (Actinobacteria) | K08384 | spoVD; stage V sporulation protein D (sporulation-specific penicillin-binding protein) |
| | Glycogen synthesis | K08822 | GSK3A; glycogen synthase kinase 3 alpha [EC:2.7.11.26] |
| | Sporulation (Actinobacteria) | K12576 | SPO12; sporulation-specific protein 12 |
| | Sporulation (Actinobacteria) | K12771 | SPA; sporulation-specific protein 1 [EC:2.7.11.1] |
| | Sporulation (Actinobacteria) | K12772 | SPD; sporulation-specific protein 4 |
| | Sporulation (Actinobacteria) | K12773 | SPR3; sporulation-regulated protein 3 |
| | Sporulation (Actinobacteria) | K12783 | SSP1; sporulation-specific protein 1 |
| | Sporulation (Actinobacteria) | K13532 | kinD; two-component system, sporulation sensor kinase D [EC:2.7.13.3] |
| | Sporulation (Actinobacteria) | K13533 | kinE; two-component system, sporulation sensor kinase E [EC:2.7.13.3] |
| | Glycogen synthesis | K16150 | K16150; glycogen synthase [EC:2.4.1.11] |
| | Exopolysaccharide synthesis | K16566 | exoY; exopolysaccharide production protein ExoY |
| | Exopolysaccharide synthesis | K16567 | exoQ; exopolysaccharide production protein ExoQ |
| | Exopolysaccharide synthesis | K16568 | exoZ; exopolysaccharide production protein ExoZ |



| | Sporulation (Actinobacteria) | K16947 | SPR28; sporulation-regulated protein 28 |
| | Glycogen synthesis | K20812 | glgA; glycogen synthase [EC:2.4.1.242] |






Table A8. Abundance (in copy number (CN)) of each patch type within each group of gene.

| Group | Patch Type | Abundance (in CN) |
|---|---|---|
| DNA conservation | Barren | 16,153.38 |
| DNA conservation | Nest | 47,287.31 |
| DNA conservation | Shrub | 46,252.92 |
| DNA conservation | Shrub&Nest | 30,860.48 |
| DNA repair and degradation | Barren | 12,091.56 |
| DNA repair and degradation | Nest | 27,516.74 |
| DNA repair and degradation | Shrub | 27,102.20 |
| DNA repair and degradation | Shrub&Nest | 20,810.48 |
| Lithotrophs | Barren | 11,856.26 |
| Lithotrophs | Nest | 73,242.15 |
| Lithotrophs | Shrub | 65,602.91 |
| Lithotrophs | Shrub&Nest | 29,183.05 |
| Nitrogen | Barren | 14,971.68 |
| Nitrogen | Nest | 29,265.84 |
| Nitrogen | Shrub | 30,326.47 |
| Nitrogen | Shrub&Nest | 25,184.32 |
| Organotrophs | Barren | 69,296.86 |
| Organotrophs | Nest | 16,1271.21 |
| Organotrophs | Shrub | 15,0159.89 |
| Organotrophs | Shrub&Nest | 90,170.34 |
| Phototrophy | Barren | 6,949.817 |
| Phototrophy | Nest | 17,722.912 |
| Phototrophy | Shrub | 19,736.83 |
| Phototrophy | Shrub&Nest | 15,555.43 |
| ROS-damage prevention | Barren | 33,660.03 |
| ROS-damage prevention | Nest | 93,064.68 |
| ROS-damage prevention | Shrub | 88,543.76 |
| ROS-damage prevention | Shrub&Nest | 60,566.25 |
| Sporulation capsule & C-storage | Barren | 2,129.44 |
| Sporulation capsule & C-storage | Nest | 14,338.20 |
| Sporulation capsule & C-storage | Shrub | 12,904.33 |
| Sporulation capsule & C-storage | Shrub&Nest | 5,514.04 |





Table A9. Chi-square values and p-values of the Dunn tests between patches done on the functional
prediction results. Bold numbers are significant (< 0.05)

| Comparisons | Nitrogen | ROS-damage | Sporulation | Phototrophy |
|---|---|---|---|---|
| **Control - Nest** | **0.0278** | **0.0046** | **0.0014** | **0.0207** |
| **Control - Shrub** | **0.0271** | **0.0212** | **0.0073** | **0.0235** |
| **Nest - Shrub** | 0.4790 | 0.2545 | 0.2623 | 0.4516 |
| **Control - Shrub&Nest** | **0.0140** | **0.0207** | **0.0421** | **0.0164** |
| **Nest - Shrub&Nest** | 0.3888 | 0.2860 | 0.1046 | 0.4625 |
| **Shrub - Shrub&Nest** | 0.3653 | 0.4693 | 0.2545 | 0.4134 |
| **Chi-square** | 6.1179803 | 7.80073892 | 10.0155172 | 6.28472906 |

| Comparisons | Organotrophy | DNA Conservation | DNA Repair | Lithotrophy |
|---|---|---|---|---|
| **Control - Nest** | 0.0513 | **0.0038** | **0.0110** | **0.0066** |
| **Control - Shrub** | 0.2267 | **0.0121** | **0.0227** | **0.0320** |
| **Nest - Shrub** | 0.1746 | 0.3077 | 0.3577 | 0.2391 |
| **Control - Shrub&Nest** | 0.2549 | **0.0060** | **0.0085** | 0.1165 |
| **Nest - Shrub&Nest** | 0.1653 | 0.4376 | 0.4625 | 0.0991 |
| **Shrub - Shrub&Nest** | 0.4725 | 0.3668 | 0.3221 | 0.2676 |
| **Chi-square** | 2.69926108 | 9.30837438 | 7.53793103 | 6.68743842 |





Table A10. Results of the adonis analysis of the impact of soil parameters on the bacterial community.

| Soil parameter | R2 | P-value |
|---|---|---|
| $NH_4^+$ | 0.03383 | 0.451 |
| pH | 0.01542 | 0.948 |
| $NO_3^-$ | 0.03141 | 0.512 |
| OM | 0.04244 | 0.263 |
| Water | 0.03851 | 0.355 |
| P | 0.03863 | 0.343 |
