# Peer review of "The role of ecosystem engineers in shaping the diversity and"

_SOIL, 2021_

## Author Response (AR1)

**Comment on soil-2021-29**
Anonymous Referee #2

Referee comment on "The role of ecosystem engineers in shaping the diversity and function of arid soil bacterial communities" by Capucine Baubin et al., SOIL Discuss., https://doi.org/10.5194/soil-2021-29-RC2, 2021

- I enjoyed reading the work presented in this manuscript. The question is straight forward and important to better understand microbial communities among patch types in arid ecosystems. The MS is written well. There are several instances in this MS where I was unsatisfied with the amount of information presented in their methods that need clarification before its clear to the reader how the study was performed. Below are specific comments regarding this issue that would ideally make the manuscript repeatable. As it stands, it is not based on the lack of information provided and significant rewrite is recommended. There are also some overall recommendations on how to present results.

  ➢ *Thank you for your comments. Additional information in the methods were added and the results were re-graphed to add your comments and to make sure the data were accurately presented.*

Methods

- How were the soils actually sampled? There is no mention of coring instrument, number of subsamples within replicate plots. Please add in this information. Line 100 states there is 5g for molecular work and 20g for water content so must have taken multiple subsamples and pooled into a composite sample.
  ➢ *The following information was added to the manuscript: "We sampled 14 random experimental blocks, from each of the four patches (4 patch types x 14 blocks = 56 samples). The samples were collected using a scoop that was sterilized between each sampling using 70% technical ethanol. Soil was collected from the top 5 cm after removal of the crust and debris. Three subsamples of ~100g were collected from each block and pooled together. In the lab, samples from two adjacent blocks were composite and homogenized using a 2 mm sieve. The samples were then separated for consecutive analyses: 15 g of each soil sample was stored in -80 °C for bacterial analysis; 25 g was used to determine the water content in the soil; and the rest was used for the measurements of physico-chemical properties.*

- Please add more information on how soil chemistry was measured. Citing standard methods and then not detailing how nitrate, ammonium, and P were extracted (what concentration of extract? How were the extracts measured – on what instrument?) is not enough information.
  ➢ *In response to the reviewer comment we have added the required information to the material and methods and it now reads:*
  *Soil physico-chemical analysis*
  *The physico-chemical parameters of the soil samples were assessed following the standard methods (SSSA, 1996). Water content was measured by gravimetry. Other parameters were measured as follows by the Gilat Hasade Services Laboratory (Moshav Gilat, Israel). The pH was measured in saturated soil extract (SSE). Phosphorus (P) was extracted by the Olsen method using a 0.5M sodium bicarbonate solution ($NaHCO_3$) and the absorbance of the final solution was measured at 880nm using a spectrophotometer. Nitrate ($NO_3^-$) and ammonium ($NH_4^+$) were extracted with a 2N potassium chloride (KCl) solution and measured at 520 nm and 660 nm,*

*respectively. Organic matter (OM) content was determined by the Walkley-Black method using a dichromate oxidation (Cr2O7-2) and the amount of oxidizable OM is measured at 600 nm.*

- Likewise to point 2, the authors must provide more details regarding their "community analysis" (I would refer to this as bioinformatics analysis and leave out the information regarding ordinations and statistics). Specifically: What parameters did you use for DADA2 including trimming or truncating sequences after reviewing sequence quality?
  - ➢ *This information was added under Bioinformatics analysis (as suggested) and it reads: "Trimming was done using TrimGalore. Briefly, all reads with a quality less than 20 and shorter than 150 bp were removed and the rest were further analysed."*

- Its assumed that the authors used a 99% identity cutoff for ASVs – can they confirm?
  - ➢ *The information was added under Bioinformatics analysis section: "A 99% identity cutoff for ASVs was used."*

- What additional steps were included after taxonomic assignment and what database classifier was used? Were contaminants (mitochondria, unclassified, chloroplasts) removed prior to downstream analysis?
  - ➢ *The information was added under Bioinformatics analysis section: "For taxonomic assignment we used Silva v132 (specifically, Silva v132 classifier). All non-bacterial data was characterized as unclassified and removed."*

- Were the sequences sub-sampled at the same sequencing depth and/or normalized to account for that?
  - ➢ *We have added this information to the Bioinformatics analysis section: "The sequences were sub-sampled at 5000 reads."*

- I recommend having a separate statistical analysis section after describing how authors measured their response variable (at end of the methods). It seems as though there are statistics that were included as supplemental material but not described at all in the manuscript (e.g., adonis test included chemistry variables). Please describe all statistics used here and include any statistical software and package used (adonis is the function, but the actual test is a permutational multivariate ANOVA).
  - ➢ *Statistics section was designated: "The statistical analysis was done using R (R Core Team, 2016). To visualize the differences between patch types, an NMDS plot was created using the Bray-Curtis dissimilarity and the significance of these differences was analysed using a non-parametric analysis of similarity (ANOSIM) ('vegan' package (Oksanen et al., 2014)). The envfit function ('vegan' package (Oksanen et al., 2014)) was applied on the NMDS data to evaluate the effect of soil parameters on the bacterial community. The NMDS was plotted using the 'ggplot2' package (Wickham, 2016) and the arrows representing the effect of each soil parameter as well as the centroids for each patch type, calculated using envfit, were added to the plot. The bacterial data were analysed using the 'phyloseq' package (McMurdie et al., 2017). All non-bacterial data have been characterized as unclassified and removed. The relative abundance, whenever higher than 0.05%, of each phylum was calculated and then plotted using a stacked bar plot ('ggplot2' package (Wickham, 2016)). The significance of difference between patch types was assessed using a non-parametric test: Kruskal-Wallis test and a post-hoc Dunn test (Dinno, 2017; Dunn, 1964; Kruskal and Wallis, 1952). All sequences retrieved in this study were uploaded to BioProject*

*(https://www.ncbi.nlm.nih.gov/bioproject) under the submission number PRJNA484096."*

Results

- L174: NMDS does not suggest any significant differences. Similar to PCA, it is just a visualization tool to view multivariate responses such as community composition. Please correct.

  ➢ *We agree. Therefore, in addition to the visual results presented by the NMDS, we included ANOSIM and clearly separated these in the text, so it is clear which is used for visualization and which for significance testing: "To visualize the differences between patch types, an NMDS plot was created using the Bray-Curtis dissimilarity and the significance of these differences was analysed using a non-parametric analysis of similarity (ANOSIM)"*

L156: I'd prefer a table of mean and standard error values for each physico-chemical variables presented along with statistical outputs. PCA and Figure 1: + or – signs are unnecessary. The direction of the arrows/vectors denotes the direction of change. Table A10 suggests either the same or a separate perMANOVA test was run with physiochemical data. This test is for categorical data. If the authors wish to understand environmental correlates, I recommend either the envfit function along with NMDS or a constrained ordination, such as redundancy analysis (or whatever appropriate analysis fits your data structure). Figure 2: Please include centroids or at least ellipses around different patch types similar to Figure 1 – this is quite helpful in delineating compositional changes.

  ➢ *The figures and table were thoroughly changed in accordance with this comment: According to the comment on Table A10, we have applied envfit analysis on the data. The results showed that most soil parameters can correlate with the barren soil community but not to the three other patch types. Due to the significance of these results a table of soil parameters was added, we replaced Figure 1 by a table of soil parameters including means and standard errors. We also replaced Figure 2 by an NMDS with the envfit vectors for the physico-chemical parameters and centroids for the different patch types.*

[Figure]

*Figure 1. Non-Metric Multidimensional Scaling (NMDS) of the soil 16S microbial communities in the dry season under different patch types. The centroids for each patch type is represented by a dashed circle. The arrow vectors represent the effect of each soil physico-chemical characteristic on the bacterial community calculated with the envfit function. NO3- = Nitrate, NH4+ = Ammonium, OM = Organic Matter content, P = Phosphorus, Water = Water content. The patch types are significantly different from each other (ANOSIM, R= 0.28247; p-value = 0.001). P, OM, NO3-, and NH4+.*

- Please separate Proteobacteria out to classes (Alpha-,Beta-,etc.) – this is important information as these classes may differ in their ecologies across ecosystems and fairly routine practice for working with 16S community data.
  - ➢ *In accordance with the comment, the Proteobacteria were separated to class level (alpha-, delta-, and gamma- proteobacteria) and Figure 3 now includes different shades of green that correspond with the different classes of the Proteobacteria.*

- What is the justification for only targeting 3 of the dominant phyla? Based on Figure 3, there are several other phyla that may be shifting among patch types, such as Bacteroidetes, Acidobacteria, and Firmicutes. These phyla, although less abundant, are important components of the community. If interested in dormancy, Firmicutes is particularly important in response to drought (see Placella et al. 2012. PNAS).
  - ➢ *Actinobacteria and Proteobacteria are the main phyla in the community, as was demonstrated in previous reports (Bachar et al., 2012, 2010; Vonshak et al., 2018). For Deinococcus-Thermus, we decided to target it due to its significant changes in its population and the important role it serves in the community (see (Baubin et al., 2019)). Following the reviewer's comment, we added additional information on the relative abundance of Chloroflexi , Bacteroidetes, and Firmicutes:*
    *Firmicutes - the relative abundance of this phylum was significantly higher in the Shrub&Nest patch than in the barren and the shrub patches*

*Bacteroidetes - the Nest patch had a significantly lower relative abundance than the other patches.*
*Chloroflexi - there was a significant decrease in relative abundance in shrub, nest and Shrub&Nest patches compared to the barren patch.*

- Also consider what other taxonomic classifications are comprising these changes in phyla among patch types – I recommend digging deeper into the taxonomic classifications (dominant families or even genera that change among patch types).
  - ➢ *Barplots of the community at the Order and the Family level was added to the Appendix. They show difference between patch types, however, the resolution is not high enough to be able to draw significant conclusions.*

Referee comment on "The role of ecosystem engineers in shaping the diversity and function of arid soil bacterial communities" by Capucine Baubin et al., SOIL Discuss., https://doi.org/10.5194/soil-2021-29-RC2, 2021

The MS represents a well-prepared experiment with clear questions and creative approach to soil microbiology. Everything is nicely described and presented but the discussion, which is not explaining the details of results well. I think that the authors should try to explain firstly why the patches have similar outcome while the effect of the two EEs are clearly different,

➢ *The patches have similar outcome because they have similar effects on the soil microbial community, i.e., changes in the water availability and physical conditions of the soil. However, it doesn't always mean that whenever the combination of EEs have the same effect is the sum of the singular effects. Rather, the combine effect implies change in the community though the contribution of each EE is not accumulative. We have mentioned that in the discussion but rewrote this part to make it clear: "The EE patches analysed in this study share the same habitat and resources but their impacts are distinct (Passarelli et al., 2014), and thus, their joint impact is non-additive. The behaviour of each EE is important as it becomes a feature of the combined impact of both EEs (Alba-Lynn and Detling, 2008). However, the effect of both EEs together cannot be inferred from their individual environmental impact or from their mutual interaction (Gilad et al., 2004)."*

secondly why the two individual EEs did not produce the effect to soil properties although they change the microbial community structure.

➢ *The two individual EEs change the physical conditions of the soil (evaporation, temperature, soil moisture retention) only during the wet season (Kidron, 2009). However, previous results have shown that the higher soil water content homogenize the microbial community (Baubin et al., 2019). Here, we were sampling in the dry season where differences between patches are pronounced (Baubin et al., 2019) as a result of the short-term effects on the soil properties during the wet season. we hypothesize that a long-term effect of the soil properties on the microbial community, which means that the observed differences in the dry season are directly linked to the changes in soil properties that happened in the wet season. To reiterate this point, we rewrote a section in the discussion: " The behaviour of each EE is important as it becomes a feature of the combined impact of both EEs (Alba-Lynn and Detling, 2008). However, the effect of both EEs together cannot be inferred from their individual environmental impact or from their mutual interaction (Gilad et al., 2004).[...] Therefore, the prolonged water availability and altered physical conditions from the wet season may hold lasting effects on the communities structure (Baubin et al., 2019), shaping the composition and functions observed here (Figure 2 and 3)."*

Also, although there were not significant differences in the functional patterns between the two patches and the combination, it seems from the graphs that the variability was much lower for the combined patches.
So, I would try to use a turnover for explaining why there is no effect on the soil properties in the individual patches and possibly resilience or resistance based on increased complexity, leading to stabilization of soil, or possibly reduced turnover for explaining the effect of combined EEs. Well, maybe you can find some other ways how to explain these patterns in more detail, but I think that the MS would benefit from more depth in those two issues. It can be just two, three sentences.

➢ *Thank you for your comments. The discussion has been modified as described above.*

More specific comments:

- L59 Better introduce the sentence starting "The community's...
  - ➤ *The sentence was changed to: "This taxonomical response to changes in the physico-chemical conditions is linked to the potential function of the community."*

- L145 Use other, more specific citation than the textbook.
  - ➤ The textbook citation was changed to Galloway et al., 2004.

- L230 Change to for metabolism-related and survival-related.
  - ➤ *Corrected.*

- L266-271 This part is not very useful, remove.
  - ➤ *Removed.*

- L271 The sentence starting "Here.." is a good beginning for the explanations. At this part you should continue to a greater detail of arguments.
  - ➤ *Additional information was added to this part: "These conditions are the results of the collective dynamics of many individuals from two drastically different species, which cannot be predicted by the individual impact of each EE (Gilad et al., 2004). And while the soil parameters might be modified by the presence of both EEs, the microbial community might take a longer time to change, due to their slow turnover in the dry season."*

- L281 I would suggest not to include the third EE for explaining the discrepancies. There are no discrepancies, just results to be explained. This paragraph leads the MS to a rather weak conclusion but why to do that if the results are strong. Similarly, the conclusions should be more elaborated to connect this study to a broader picture of microbial functional distribution in dry habitats.
  - ➤ *The part about the third EE was removed and the section rewritten: "In this study, we focused on two EEs only, but there are many EEs in one ecosystem and knowing their joint impact would help explain the nutrient turnover and the bacterial communities in this ecosystem."*

References

Bachar, A., Al-Ashhab, A., Soares, M. I. M., Sklarz, M. Y., Angel, R., Ungar, E. D., and Gillor, O.: Soil Microbial Abundance and Diversity Along a Low Precipitation Gradient, Microb Ecol, 60, 453–461, https://doi.org/10.1007/s00248-010-9727-1, 2010.

Bachar, A., Soares, M. I., and Gillor, O.: The effect of resource islands on abundance and diversity of bacteria in arid soils, 63, 694–700, https://doi.org/10.1007/s00248-011-9957-x, 2012.

Baubin, C., Farrell, A. M., Šťovíček, A., Ghazaryan, L., Giladi, I., and Gillor, O.: Seasonal and spatial variability in total and active bacterial communities from desert soil, Pedobiologia, 74, 7–14, https://doi.org/10.1016/j.pedobi.2019.02.001, 2019.

Kidron, G. J.: The effect of shrub canopy upon surface temperatures and evaporation in the Negev Desert, Earth Surf. Process. Landforms, 34, 123–132, https://doi.org/10.1002/esp.1706, 2009.

SSSA, S. S. S. of A.: Methods of Soil Analysis: Part 3 Chemical methods, 5.3, edited by: Sparks, D. L., Page, A. L., A, H. P., Loeppert, R. H., Soltanpour, P. N., Tabatabai, M. A., Johnson, C. T., and E, S. M., 1390 pp., 1996.

Vonshak, A., Sklarz, M. Y., Hirsch, A. M., and Gillor, O.: Perennials but not slope aspect affect the diversity of soil bacterial communities in the northern Negev Desert, Israel, 56, 123, https://doi.org/10.1071/SR17010, 2018.